# Rhodanese-Fold Containing Proteins in Humans: Not Just Key Players in Sulfur Trafficking

**DOI:** 10.3390/antiox12040843

**Published:** 2023-03-31

**Authors:** Razan Alsohaibani, Anne-Lise Claudel, Romain Perchat-Varlet, Séverine Boutserin, François Talfournier, Sandrine Boschi-Muller, Benjamin Selles

**Affiliations:** IMoPA, CNRS, Université de Lorraine, F-54000 Nancy, France

**Keywords:** Rhodanese-fold, sulfur trafficking, cysteine persulfide, promiscuous activities, MoCo maturation, tRNA thiolation

## Abstract

The Rhodanese-fold is a ubiquitous structural domain present in various protein subfamilies associated with different physiological functions or pathophysiological conditions in humans. Proteins harboring a Rhodanese domain are diverse in terms of domain architecture, with some representatives exhibiting one or several Rhodanese domains, fused or not to other structural domains. The most famous Rhodanese domains are catalytically active, thanks to an active-site loop containing an essential cysteine residue which allows for catalyzing sulfur transfer reactions involved in sulfur trafficking, hydrogen sulfide metabolism, biosynthesis of molybdenum cofactor, thio-modification of tRNAs or protein urmylation. In addition, they also catalyse phosphatase reactions linked to cell cycle regulation, and recent advances proposed a new role into tRNA hydroxylation, illustrating the catalytic versatility of Rhodanese domain. To date, no exhaustive analysis of Rhodanese containing protein equipment from humans is available. In this review, we focus on structural and biochemical properties of human-active Rhodanese-containing proteins, in order to provide a picture of their established or putative key roles in many essential biological functions.

## 1. Introduction

Proteins harboring a Rhodanese (Rhd) domain are ubiquitous and are involved in a variety of physiological processes, from metabolic pathways and signaling to protein posttranslational modifications [1,2,3,4]. The role of Rhd-containing proteins in most of these pathways resides in the capability of such functional protein domains to catalyze sulfur atom transfers from a donor molecule to an acceptor one, thus, participating in sulfur atom trafficking. In particular, they are now recognized as key players in the emerging gasotransmitter hydrogen sulfide (H_2_S) homeostasis in humans. H_2_S is described as an ubiquitous and pleiotropic molecule with a broad spectrum of targets, including nuclear factor-kappa (NF-κB), p38 mitogen-activated kinase (p38 MAPK), nuclear factor erythroid 2-related factor 2 (Nrf2), p53 and mTOR, the target of rapamycin [5,6,7,8]. Based on its chemical properties, the reactions involving H_2_S are categorized into the following three groups: 1—reactions with metal centers, 2—cross-talks with ROS and RNS, 3—protein persulfide formation [9]. More recently, Rhd-containing proteins were reported to be involved in tRNA thiolation (i.e., post-transcriptional modifications), stabilizing tRNA tertiary structure and/or permitting fine-tune decoding processes [10], or in maturation of the molybdenum cofactor (MoCo), a prosthetic group present in essential proteins, such as sulfite oxidase (SO), xanthine oxidoreductase (XOR), and aldehyde oxidase (AO) [11].

Hence, understanding the Rhd structural domain landscape is an essential step to understanding the importance of such proteins in human physiology. Here we summarize the current understanding on active human Rhd proteins, in both mechanistic and physiological terms.

## 2. The Rhodanese-Fold and Domains Organization

The so called “rhodanese” fold is an α/β structural arrangement, containing around 100 residues, found in a broad range of protein structures, encoded by prokaryotes and eukaryotes genomes (Figure 1a) [1,2].

Despite a conserved overall structural arrangement, primary structures adopting the Rhd fold are highly variable, making global sequence comparison between functionally or evolutionary distant Rhd-containing proteins difficult. Meanwhile, the growing number of available Rhd 3D structures from a variety of proteins (215 entries at RCSB database, https://www.rcsb.org accessed on 5 September 2022) allows a direct inspection of Rhd-domain conserved structural signatures. As an example, an in silico large scale identification and analysis of Rossmann-like structural domains revealed that the Rhd fold belongs to this vast family. Its structure is composed of a three-layered α/β sandwich (a characteristic of Rossmann-like protein) in which, the central parallel β sheet is surrounded by 4 to 5 α-helices (Figure 1a) [12]. Authors also identified two highly conserved structural motifs within the Rhd domain, at the end of the β1 strand, corresponding to a DxR motif of unknown function, and the end of the β3 strand corresponding to a (C/D)x_4/5_(R/T) loop motif (Figure 1b, Appendix A) [1,3]. Quite interestingly, despite some sequence variability, these two structural motifs are conserved within Rossmann-like subfamilies, revealing a promiscuous fold adapted to perform various functions. Thereby, Rhd domains can be defined as catalytically active or inactive, depending on the presence or absence, respectively, of a cysteine residue within the loop located at the end of the β3 strand (Figure 1a) [1]. For the active ones, the so-called catalytic loop is made of a Cx_4_R/T or Cx_5_R motif that supports various chemical reactions. By contrast, catalytically inactive Rhd domains are rather considered as protein–protein interaction mediators [13].

Rhd domain-containing proteins also exhibit variable domain organizations, with one or several Rhd domains fused or not to other functional domains [1,2,3,14,15]. Considering the catalytically active/inactive criteria previously defined, the following four classes can be defined (Figure 1c): (1) mono-domain proteins, with a unique active Rhd domain; (2) two-domain proteins, with an inactive and an active Rhd domains; (3) an active Rhd domain fused to unrelated domains; (4) an inactive Rhd domain fused to unrelated domains.

## 3. Reactions Catalyzed by Active Rhodanese Domains

The active Rhd domains, bearing a cysteine residue in the catalytic loop, are able to catalyze several classical reactions as follows: a sulfur transfer for sulfurtransferases, a phosphate group transfer for phosphatases, or an arsenate reduction for arsenate reductases. The latter two were reported for the CDC25 proteins [16,17].

Sulfurtransferases catalyze the transfer of a sulfur atom from a donor substrate to an acceptor one. Depending on their preferred donor substrate in vitro, these enzymes can be classified into the following two groups: thiosulfate sulfurtransferases (TSTs) for thiosulfate, and 3-mercaptopyruvate sulfurtransferases (3MSTs or MPSTs) for 3-mercaptopyruvate [2]. The two-step reaction is initiated by the nucleophilic attack of the catalytic cysteine on the sulfur atom of the substrate, leading to the formation of a persulfide intermediate. Then, the sulfur atom is transferred to an acceptor to form the product and regenerate the cysteine under thiolate form (Figure 2a) [18]. The two-step mechanism for the phosphatase-catalyzed reaction [19] starts with the nucleophilic attack of the catalytic cysteine on the substrate phosphate group to form a phosphoenzyme intermediate that is further hydrolyzed (Figure 2b). Finally, arsenate reductases catalyze the two-electron reduction in arsenate (As(+V)) to arsenite (As(+III)), and the proposed mechanism involves the nucleophilic attack of the catalytic cysteine on the arsenic atom of the substrate to form a thioarsenate intermediate, which is then reduced to arsenite by glutathione and glutaredoxin (Figure 2c) [20,21].

## 4. The Human Rhodanese Repertoire

Because of their highly divergent nature, sequence identity search programs cannot be used to exhaustively identify Rhd domains from sequence databases. A keyword-based search of the human Rhd domains present in the UniProt library [22] identified 30 proteins (Appendix A), 23 of which possess a Rhd domain (the 7 remaining are described as interacting with Rhd-containing proteins). Structural information is available for these 23 proteins, either PDB structures for 10 of them, or 3D models in the Alphafold database for all [23]. A thorough analysis of these structures or models shows that 8 of them possess a cysteine-containing catalytic loop and thus should be active (MPST, TST, TSTD1, CDC25A-B-C, MOCS3 and TSTD2) (Figure 3a). Interestingly, superimposing all the catalytic loops reveals noticeable structural homology, even for CDC25-type proteins containing a loop that is one residue longer (Figure 3b). Despite this structural homology, it was proposed by several studies that catalysis features and substrate specificities of the Rhd domain-containing protein are mostly driven by the residue composition of the catalytic loop [1,24,25].

Among the eight active Rhd-containing proteins, four catalyze a sulfur transfer reaction (MPST, TST, TSTD1, MOCS3) and three catalyze phosphatase reactions (CDC25A/B/C), whereas no typical Rhd-dependent activity is proposed for the last one (TSTD2). In terms of domain organization, TSTD1 belongs to class one, MPST and TST to class two, and CDCD25A/B/C, MOCS3 and TSTD2 to class three (Figure 3c). These 8 proteins were characterized in vitro and/or *in cellulo*, except for TSTD2 (Figure 4).

## 5. TSTD1: The Unique Class One Rhd Enzyme, So Far

TSTD1 is encoded by the *KAT* gene located on chromosome 1 (1q23.3), which is 1.6 kilobases long and consists of 4 exons and 3 introns [30]. In vivo, three transcripts are generated from *KAT* by alternative splicing, thus leading to the production of 3 TSTD1 isoforms (1-1, 1-2, 1-3) (Figure 3c). The TSTD1-1 isoform (115 residues and 12,530 Da) is issued from the longest transcript, containing the four exons. The TSTD1-2 isoform (74 residues and 8274 Da) that is truncated at its N-terminus compared to TSTD1-1 corresponds to the shortest transcript, lacking exon 2. The TSTD1-3 isoform (109 residues and 11,750 Da) with a truncated C-terminal end and a decapeptide not present in TSTD1-1 and 1-2 (decapeptide encoded by intron 3) is produced from the medium transcript, lacking exon 4 and containing intron 3.

The sulfurtransferase activity of TSTD1 isoforms has been studied in vitro [35,36]. First, Melideo and colleagues have investigated potential sulfurtransferase activities of all three TSTD1 isoforms, with thiosulfate as donor and glutathione as acceptor, and they have demonstrated that TSTD1-1 is the only isoform displaying sulfurtransferase activity under such conditions [35]. In addition, Libiad and co-workers have investigated the thiosulfate sulfurtransferase activity of TSTD1-1, with various sulfur acceptor substrates (Figure 5b) [36]. In these conditions, the TSTD1-1 protein exhibits affinity for KCN, GSH, Cys or homocysteine (HCys) in the mM range, but a *K*_M_ for thioredoxin around 17 µM, a value close to those reported for other physiological thioredoxin partners [36,37,38,39]. This suggests that, in vitro, human thioredoxin is the final sulfur acceptor for TSTD1.

In addition, Melideo and colleagues revealed a potential mechanistic crosstalk between TSTD1 and a protein with Persulfide Dioxygenase (PDO) activity. Indeed, they demonstrated that the presence of PDO accelerated the TSTD1 catalysis toward thiosulfate by consuming the TSTD1 reaction product (GSSH). Moreover, the fact that fusions between PDO and Rhd-fold protein domains exist proposes that TSTD1 protein is part of the mitochondrial sulfide oxidation unit complex (SOU complex), the main actor of H_2_S detoxication in mammalian cells. Meanwhile, no in cellulo data supports this hypothesis so far.

A high resolution crystal structure of the TSTD1-1 isoform was recently reported (Figure 5a) [36]. TSTD1-1 possesses a canonical Rhd fold composed of five stranded β-sheet surrounded by six α-helices. Authors also identified a potential specific feature of TSTD1 protein compared to multidomain sulfurtransferases, because the catalytic cysteine lies in a shallow pocket, whereas it is more deeply buried for MPST or TST proteins.

In humans, *TSTD1* transcripts accumulate in different tissues, such as the kidney, liver, skeletal muscle, heart, colon, thymus, spleen, placenta, or lung [30]. In addition, TSTD1 proteins accumulate in various cancer derived cell lines [30]. Because the human colon is physiologically exposed to high levels of thiosulfate due to gut microbiome activities, Libiad and colleagues recently, investigated the TSTD1 protein localization within this tissue [36,40]. They demonstrated that TSTD1 accumulates within the luminal face of the colon, the potential major site of interaction between the microbiome and human cells.

## 6. The Two Rhd Classes, TST and MPST (Two Evolutionarily Related Sulfurtransferases with Different Functions) 

### 6.1. TST

The human TST is also called Rhodanese in the literature, due to the name of the first characterized TST protein denominated as RhoBov (Bovin Rhodanese) [41,42,43]. The *TST* gene is located on chromosome 22 (22q12.3), and the protein (297 residues and 33 kDa) is produced from a unique 1.3 kb long mRNA [44]. Regarding 3D-structure, no information is available for the human enzyme, except for a model predicted by Alphafold (https://alphafold.ebi.ac.uk/entry/Q16762, accessed on 18 October 2022). Nevertheless, the 3D-structure of the RhoBov homolog protein, with 89% sequence identity, is available in the PDB (PDB 1BOH). As previously mentioned, the TST protein is composed of two Rhd domains with a unique catalytic cysteine residue located in the C-terminal one (Figure 3c and Figure 6a). The active site is walled in by residues from both domains and resides in the interdomain cleft. The catalytically active cysteine residue is embedded in the CRKGVT loop, whose sequence clearly differs from those of other sulfurtransferase family members [1]. In vitro, TST efficiently catalyzes the sulfur transfer from thiosulfate to cyanide to generate thiocyanate and sulfite (Figure 6b). This reaction explains the rationale for using thiosulfate as an antidote against cyanide poisoning. It has also been shown that TST has a 40-fold higher catalytic efficiency in the presence of glutathione-persulfide (GSSH) compared to thiosulfate (Figure 6c) [45]. In addition, sulfite is a rather good sulfur-acceptor substrate, which leads to the formation of thiosulfate. Altogether, formation of thiosulfate from sulfite and GSSH is much more efficient than the reverse reaction, with at least 400-fold higher *k*_cat_/*K*_M_ value (Figure 6b,c), suggesting that TST preferentially produces, rather than utilizes, thiosulfate. Two other thiol acceptors, cysteine and homocysteine, showed higher efficiency than GSH, primarily due to higher *k*_cat_ values. However, one needs to keep in mind that steady-state kinetics are not totally appropriate to define the substrate specificities of this enzyme, since TST exhibited kinetic cooperativity with respect to many substrates. This may result from unexplained hysteresis behaviour and thus prevents relevant comparison of catalytic efficiencies [46].

Although the TST protein is targeted to mitochondria, it does not contain a N-terminal mitochondrial targeting sequence (Figure 3c and Figure 4) [28,42,47]. Interestingly, investigation of 5S RNA mitochondrial targeting revealed that two protein components were essential to this process, the TST and the MRP-L18 proteins [28,29]. Mixed cellular and molecular experiments showed that a TST/5 S RNA complex is indeed formed, mediated by the 5S RNA γ-domain, and that the MRP-L18 protein is essential for the γ-domain conformational changes that are required to form this complex [28,29]. Meanwhile, the TST protein is successfully imported into mitochondria in the absence of 5S RNA, at least in vitro, indicating that the molecular mechanism driving this process is still elusive [28]. This protein is also proposed to be involved in the mitochondrial sulfide oxidation pathway. This pathway begins with sulfide quinone oxidoreductase (SQOR) and includes a sulfur dioxygenase (also known as ETHE1 or persulfide dioxygenase, PDO) and TST. Indeed, combined kinetic data and simulations at physiologically-relevant substrate concentrations shows that SQOR predominantly catalyses the synthesis of GSSH, a substrate for TST [45,48,49].

The TST protein is abundant in the liver, kidney and colon, where it is important in the detoxification of H_2_S produced by sulfate-reducing microbiota [46]. A role of TST in the regulation of fat mass has also been proposed. This assumption is supported by the fact that expression of the *TST* mRNA in adipose tissue is positively correlated with insulin sensitivity and negatively with fat mass. Thus, the *TST* gene as a beneficial regulator of mitochondrial adipocyte function could have therapeutic significance for people with type 2 diabetes [50].

Finally, a deficiency in TST activity is associated with Leber’s disease, a rare hereditary pathology causing atrophy of the eye [51]. The study of liver biopsies of patients showed that TST activity decreases during the disease progression. Indeed, infusion of these patients with thiosulfate does not cause an increase in the urinary concentration of thiocyanate, in contrast to what is observed for healthy individuals [52]. This suggests that cyanide detoxification is suboptimal in patients with Leber’s disease, probably due to decreased TST activity.

### 6.2. MPST 

MPST, now well recognized as one of the enzymes involved in the generation of hydrogen sulfide (H_2_S), has already been subject to a recent review focused on mechanistic and physiological aspects [53].

As expected, the reported crystal structure of MPST contains two Rhd domains as follows: an inactive N-terminal domain (residues 1–138) and an active C-terminal domain (residues 165–285), connected by a long linker (residues 139–164) that wraps around and interacts tightly with both domains (Figure 3c and Figure 7a) [54]. A cleft located between the two domains houses the active site occupied by the complex formed after the first step of sulfur transfer. The catalytic cysteine persulfide (Cys_248_-SSH) located in a CGSGVT loop and the pyruvate (Figure 7a) and both domains contribute residues that line the active site. 

MPST transfers a sulfur atom from 3-MP to various physiological or artificial acceptors and, if the latter is a thiol compound, the decomposition of the product formed leads to H_2_S production (Figure 7b) [2,55,56]. Non-physiological reductants that can liberate H_2_S include DTT and 2-mercaptoethanol, whereas physiologically relevant ones are GSH, homocysteine, cysteine, dihydrolipoic acid (DHLA) and thioredoxin [25,54]. Based on steady-state kinetic parameters, thioredoxin is likely to be the better acceptor for MPST, as supported by a catalytic efficiency that is much higher than those obtained for the non-physiologically reductant DTT (*k*_cat_/*K*_M_ 400-fold lower) or the small biologically-relevant reductant GSH (*k*_cat_/*K*_M_ 52,000-fold lower) (Figure 7b). It is important to note that MPST displays non Michaelis-Menten kinetics, such as TST. This behaviour, that may result from hysteresis, could be explained by the requirement for conformational transitions to occur between the two steps of sulfur transfer (Lec et al., 2018). Moreover, MPST is the only Rhd-containing sulfurtransferase for which the persulfide formation mechanism has been deciphered [25,53]. This step is very efficient and critically depends on the electrostatic contribution provided by the CGSGVT catalytic loop. Furthermore, the water-mediated protonation of the pyruvate enolate and the S^0^ transfer from the deprotonated 3-MP to the thiolate form of the catalytic cysteine occur concomitantly.

In the human genome, only one gene located on chromosome 22 (22q12.3) encodes a MPST (also known as TUM1 or 3-MST). The cytoplasmic and mitochondrial localization of human MPST is supported by immunofluorescence studies performed in human HEK293 cells (Figure 4) [27]. This dual subcellular localization is due to the existence of two MPST isoforms, resulting from an alternative translation initiation start. The shorter isoform possesses an accessible mitochondrion targeting sequence, whereas this sequence is masked by an additional N-terminal peptide of 20 residues for the longer one [27]. These features explain the strict cytosolic or the dual cytosolic–mitochondrial localization of the long or the short form, respectively (Figure 3c and Figure 4). It should be noted that these two isoforms exhibit identical kinetic and physicochemical properties [27,57]. Moreover, the cytosolic isoform of MPST is able to interact with the cytosolic forms of the NFS1 cysteine desulfurase and MOCS3 sulfurtransferase, while the second one interacts with MOCS3 in the cytosol and NFS1 within mitochondria [27]. These results suggest additional roles for MPST depending on its subcellular location (thiolation of mitochondrial tRNAs for the MPST/NFS1complex and biosynthesis of molybdenum cofactors for the cytosolic MPST/MOCS3 complex).

MPST is involved in H_2_S production in the brain, kidney, and liver at the mitochondrial level [58]. From a pathophysiological point of view, an MPST deficiency in humans is associated to a rare syndrome called mercaptolactate-cysteine disulfiduria, characterized by mental disorder and a specific form of disulfiduria [59,60,61].

## 7. The Class Three Rhd Enzymes in Humans, the Most Versatile Rhd Domains

This class is the most divergent in terms of domain nature and organization, catalytic activity and physiological function, with three enzymes, among which, one was recently discovered.

### 7.1. CDC25s

CDC25 (Cell Division Cycle 25) phosphatases belong to the family of dual-specificity phosphatases, i.e., they are able to dephosphorylate both the pThr and pTyr residues of their substrates, the cyclin-dependent kinases (CDK) [62].

The CDC25 family in humans is composed of three proteins (CDC25A, CDC25B and CDC25C) encoded by three different genes localized in chromosomes 3, 20 and 5, (3p21.31, 20p13 and 5q31.2), respectively. CDC25 genes produce transcripts that are subjected to alternative splicing, leading to the production of two variants for CDC25A and four variants for CDC25B or CDC25C (Figure 3c). Canonical representatives for CDC25s isoforms are 524, 580 and 473 residues, which are proteins for CDC25A, CDC25B and CDC25C, respectively (Figure 3c) [63,64,65,66].

The structure of CDC25s can be divided into two main domains, the extremely divergent regulatory N-terminal domain and the catalytic C-terminal Rhd domain (Figure 3c) [67]. To date, available 3D structures for CDC25s only concern their respective Rhd domain (Figure 8a). The catalytic Cys484 residue, included into the HCX_5_R loop sequence of the Rhd domain, is located in a binding pocket for the substrate’s phosphate group, whereas other key elements for substrate recognition are located about 20–30 Å from the active site [68,69]. The N-terminal regulatory domain contains phosphorylation and ubiquitination sites that regulate phosphatase activity, and there are large differences between the three isoforms (sequence similarity is less than 25% and they vary in length). Moreover, this domain contains other sequence motifs, which determine the subcellular localization of the CDC25s (Figure 4) [70].

The catalytic mechanism of phosphatase activity is a classical two-step mechanism (Figure 2b) [71]. In the Michaelian complex, the phosphate group of the substrate is probably stabilized by hydrogen bonds, with the arginine and X residues of the HCx_5_R motif. Although Glu474 was recently proposed for CDC25B, the residue involved in acid/base catalysis for leaving group protonation and water molecule activation remains to be identified. Moreover, arsenate reductase activity was revealed for the catalytic domain of the human CDC25B and CDC25C, as these two enzymes were shown to catalyze in vitro inorganic arsenate reduction in the presence of glutathion and glutaredoxin (Figure 8b) [17]. Substituting alanine for the cysteine and arginine residues of the HCx_5_R catalytic motif led to the loss of both reductase and phosphatase activities, suggesting that the Rhd catalytic domain shares the same active site for dephosphorylation and arsenate reduction. CDC25s serve as key activators of the Cdk/cyclins activity during normal cell cycle progression [62,72]. CDC25A controls both the G1-to-S and G2-to-M transitions, whereas CDC25B and CDC25C regulate the G2-to-M transition (Figure 8c). The CDC25s also play an important role in the checkpoint response that prevents Cdk/cyclin activation following DNA damage [73,74]. CDC25A is mainly localized in the nucleus (Figure 4) [75]. CDC25B, the “initiator” phosphatase, is located in the nucleus during interphase, but shuttles back and forth between the cytoplasm and nucleus during the entire cell cycle. It translocates to the cytoplasm during the G2 phase, activates the cyclin B1/CDK1 complex, and then re-enters the nucleus to initiate mitosis (Figure 4 and Figure 8c) [70,76]. Moreover, CDC25s are cell cycle control proteins whose overexpression is frequently associated with a wide variety of cancers. The downregulation of CDC25C induces cell cycle arrest in G2/M phase in response to DNA damage via p53-mediated signal transduction, and its abnormal expression is associated with cancer initiation, development, metastasis and occurrence. Indeed, many studies revealed that CDC25C is highly expressed in lung, liver, gastric, bladder, prostate, esophageal and colorectal cancers and in acute myeloid leukemia, correlated to poor prognosis and low survival rates [77].

### 7.2. MOCS3

The human MOCS3 protein is encoded by the *MOCS3* gene composed of a unique 1383 bp long exon, located on chromosome 20 (20q13.13). MOCS3 is a protein of 461 residues (49 kDa), constituted of two structural domains (Figure 3c and Figure 9a) [24], a Rhd C-terminal domain and a N-terminal domain, exhibiting similarity with bacterial ThiF/MoeB/HesA family or E1 ubiquitin-like activating enzymes [1,78]. Little information is available regarding MOCS3 3D arrangement, with only one structure for the Rhd domain of the protein (Figure 9b) (PDB 3I2V). An Alphafold prediction is also available (https://alphafold.ebi.ac.uk/entry/O95396, accessed on 17 October 2022) and, as expected from the structure of MoeB homologs, the human Moeb-like domain is constituted of a central height strands β-sheet, surrounded by height helices [78]. From the 316 sequences retrieved by blast-P from Uniprot database with human MOCS3, Arabidopsis MOCS3 (CNX5/STR13) or yeast MOCS3 (also known as Uba4p) as templates, we searched whether conserved residues or sequence motifs are conserved in the two domains throughout distant sequences (Figure 9a, Appendix A). Regarding the MOCS3 Rhd domain, the DxR structural motif is strictly conserved, and the catalytic loop displays a CxxGxD/R consensus sequence. In MoeB/ThiF domain, di-cysteines motifs involved in zinc cation coordination (Cys_222_XXCys_225_ and Cys_297_XXCys_300_), reported to be essential for the folding of the protein, are conserved in addition to a cysteine residue (Cys_239_), proposed to be essential for MOCS3 function.

In vitro, isolated MOCS3-Rhd was shown to present a low sulfurtransferase activity from thiosulfate to cyanide or DTT (Figure 9c) [24,79]. Indeed, the MOCS3-Rhd thiosulfate-dependent activity is 1000-fold lower, relative to bovine TST/Rhodanese, suggesting that thiosulfate is not the MOCS3 physiological substrate (Figure 9d). This could be linked with the particular catalytic loop sequence, as suggested by a 36- to 165-fold increase in sulfurtransferase activities when MOCS3 catalytic loop was substituted by the CRKGVT sequence issued from bovine Rhodanese [79]. Interestingly, the fact that NFS1 L-Cysteine desulfurase and MPST were shown to interact both in vitro with MOCS3-RhD and in cellulo with MOCS3 suggests that they could be its physiological donor substrates (Figure 9d) [27,80]. Thus, MOCS3 is likely the first human Rhd-containing enzyme involved in protein to protein sulfur atom trafficking. 

As isolated proteins, MoeB/ThiF members catalyse ATP-dependent adenylation of C-terminal Glycine residue from sulfur carrier proteins [81]. In the context of the MOCS3 protein, both MoeB/ThiF and Rhd domains cooperate sequentially to catalyse the C-terminal thiocarboxylation of partner proteins, e.g., MOCS2 (Molybdenum cofactor synthesis 2) or URM1 (Ubiquitin-related modifier 1) that are involved in molybdenum cofactor maturation, tRNA thiolation, or proteins post-translational modification, such as peroxiredoxin urmylation (Figure 10) [26,82]. From studies of human MOCS3 and homologs, it was proposed that MOCS3 activates MOCS2 in the presence of ATP by formation of a thioester bond between conserved C239 of the MoeB/ThiF domain and MOCS2 C-terminal glycinyl residue. This covalent linkage is proposed to be essential for MOCS2 correct orientation relative to the Rhd domain during the catalysis, which consists of the nucleophilic attack of the persulfidated MOCS3-Rhd on the thiocarboxylated MOCS2 to form an acyl persulfide intermediate. Finally, release of the MOCS2 thiocarboxylate requires an unknown electron provider [24,83,84].

In human cells, the MOCS3 protein accumulates into the cytoplasm and participates in sulfur atom trafficking through two distinct pathways, tRNA thiolation and molybdenum cofactor maturation [24,26]. At the physiological level, molybdenum cofactor (MoCo) maturation pathway defect induces severe pathological conditions at the neonatal period, causing neurological abnormality, developmental delay, skeletal changes or feeding difficulties [11]. These symptoms are proposed to be due to inactivation of MoCo cofactor containing essential proteins, i.e., sulfite oxidase, xanthine oxidoreductase, or aldehyde oxidase, leading to the accumulation of toxic metabolites, such as sulfite, xanthine or hypoxanthine. MoCo deficiency symptoms were reported to be caused by mutation in *MOCS1* (Molybdenum cofactor synthesis 1), *MOCS2*, or *GPHN* (Gephyrin) genes [85]. In most cases, patients died at young age. Meanwhile, recent studies reported severe MoCo deficiency symptoms for two patients with mutation into *MOCS3* gene and highlighted the physiological importance of MOCS3 protein for MoCo maturation [86,87]. In the first case, the seventeen-year old patient presented intellectual disabilities, autism and dysmorphic features [86]. The second case was a newborn female who presented early microcephaly, hypotonia and hyper-reflexia [87]. In the two cases, biochemical/metabolites investigations related to MoCo deficiency revealed sulfocysteine, thiosulfate and xanthine, or uric acid and sulfite abnormal accumulation, respectively [86,87]. Moreover, HEK293T in cellulo investigation conducted on homozygote *MOCS3* mutant, generated by CRISPR-Cas9, offers the most complete picture of MOCS3 cellular functions [88]. First, this investigation reported a cell viability reduction as a consequence of *MOCS3* (^−^/^−^). Next, regarding MoCo maturation, it was demonstrated that sulfite oxidase (90% decrease) or aconitase (35% decrease) activity was impacted by *MOCS3* depletion [88]. Finally, authors also demonstrated the essential role of MOCS3 in tRNA post-transcriptional modification, with an accumulation of mcm^5^U-modified tRNA, the precursor of mcm^5^s^2^U tRNA molecules [88].

### 7.3. TSTD2

The human Thiosulfate SulfurTransferase-like Domain containing 2 isoform 1 (TSTD2) is encoded by the *TSTD2* gene composed of 33,289 nucleotides, including 10 exons. The gene is localized on chromosome 9 (9q22.33). *TSTD2* orthologs are widely distributed genes present in bacteria and eukaryotes, but are absent in Archaea and obligate anaerobic bacteria [89].

TSTD2 protein contains 516 residues and consists of two known domains, including a N-terminal acylphosphatase domain (UPF0176/IPR001792) and a C-terminal Rhd domain (Figure 11), but the alternative translation initiation start generates a shorter isoform of the TSTD2 protein that contains 167 residues and only the Rhd domain (Figure 3c). A model of the three-dimensional organization of the “full-length” TSTD2 protein generated from the 3D-structure of the *Legionella pneumophila* orthologous protein (PDB 4F67) is available in AlphaFold (AF-QT7W7) (Figure 11) [23]. The Rhd domain contains the canonical Rhd CX_4_R catalytic motif (residues 355-400) localised on a loop (Figure 3 and Figure 11).

Regarding functions or biochemical properties of TSTD2, no information is available in the literature. Nevertheless, another orthologous protein from *E. coli*, named YceA or TrhO, was recently characterized [89]. The functional analysis of this protein in *E. coli* has highlighted its role in the hydroxylation of tRNAs, which initiates the cmo5U synthesis and improves decoding of codons ending in G (Figure 11) [89]. The functional complementation of an *E. coli* mutant by variants of the TrhO protein underlined the role of several residues belonging to the Acylphosphatase domain (K112) or to the Rhd catalytic loop (C200, T201, G203, R205), in the production of oxygen-dependent ^ho5^U34 hydroxylated tRNAs. However, the mechanism whereby catalysis occurs remains to be elucidated. TrhO is specific for a sub-set of tRNA (Ser1, Val1, Ala1 or Thr4), and tRNAs recognition could be achieved by the positively charged β-sheet present in the acylphosphatase domain near the Rhd active-site loop. The acylphosphatase domain is also described in the InterPro databank as catalysing the hydrolysis of acyl-phosphate groups to release a carboxylic group and a phosphate molecule, but no biochemical data support this assumption. Finally, no data indicate whether the TSTD2 Rhd domain is able to catalyse classical sulfur atom transfers, dephosphorylation reactions or arsenate reduction.

At the physiological level, the cellular functions of TSTD2 protein are still unknown. TSTD2’s gene expression or protein accumulation variations were reported for human traits or diseases resistance/susceptibility, but, to date, this information is too preliminary to propose physiological implications [90,91,92].

## 8. Concluding Remarks

The increasing number of identified and characterized human Rhd domain-containing proteins supports the notion that the Rhd structural domain has evolved, as isolated protein or in combination with other domains, to fulfill different cellular functions. Although, the fine characterization of the active Rhd domains is still elusive, especially for the last one reported, the emerging picture in this catalytic promiscuity stressed the crucial role of the active-site loop, the structural domains organization and interaction networks, depending on subcellular localization. In the future, the exhaustive characterization of human Rhd-containing proteins at the biochemical and cellular level is a decisive step to better define the functions of Rhd-containing proteins, and to reveal their key contributions in human physio- and pathophysiology.

## Figures and Tables

**Figure 1 antioxidants-12-00843-f001:**
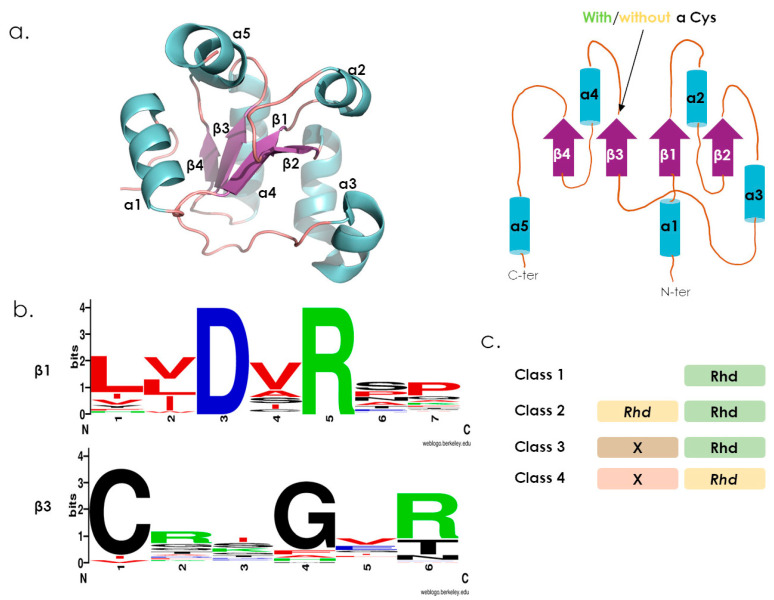
Rhodanese-fold features of Rhd-containing proteins. (**a**) 3D structure of TSTD1 (PDB 6BEV), with its topological representation. (**b**) WebLogo-generated representation (https://weblogo.berkeley.edu/logo.cgi, accessed on 07 October 2022) of the patterns issued from a multiple alignment of 50 sequences (identified by Blast with TSTD1 as template, UniProt, https://www.uniprot.org/, accessed on 20 September 2022) that cover the two structural regions located at the end of β-1 and β-3 strands in Rossmann-like fold. (**c**) The different subclasses of Rhd domain-containing proteins. Active Rhd domains are represented in green, inactive Rhd domains are in yellow, other structural domains (X) are represented in brown and pink.

**Figure 2 antioxidants-12-00843-f002:**
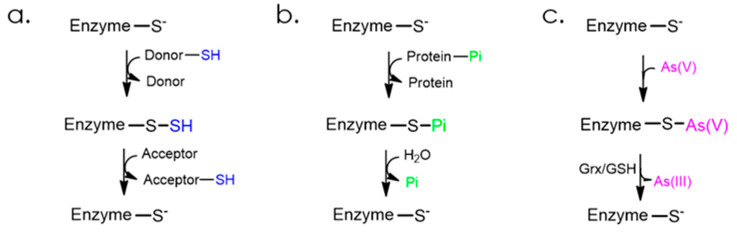
Two-step mechanisms for the reactions catalyzed by (**a**) sulfurtransferases, (**b**) phosphatases, and (**c**) arsenate reductases.

**Figure 3 antioxidants-12-00843-f003:**
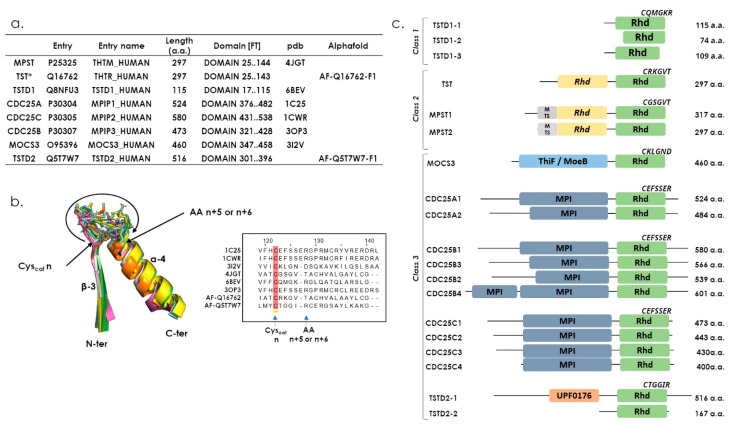
Human-active Rhd domain-containing proteins. (**a**) Eight human Rhd domain-containing proteins retrieved from the Uniprot database (https://www.uniprot.org/, accessed on 15 September 2022), indicating the length of the proteins and the position of the Rhd domain within the sequences. 3D-structures available in PDB, experimentally resolved or predicted structures by alphafold (https://alphafold.ebi.ac.uk/ accessed on 18 October 2022) are indicated. TST* is also known as Rhodanese. (**b**) Superimposition of active site loops (in sticks) flanked by the β-3 strand and the α-4 helix was made using CCP4MG software (https://www.ccp4.ac.uk/MG/ accessed on 5 July 2022) and is represented using Pymol (https://pymol.org/2/, version 2.4.1, accessed on 5 July 2002). Sequence alignment extracted from structure superimposition is represented in the right panel. (**c**) Modular organization of human-active Rhd domain-containing proteins and of their described isoforms. Active site loop sequences are indicated, and protein domains were defined using InterPro (https://www.ebi.ac.uk/interpro/, accessed on 25 September 2022).

**Figure 4 antioxidants-12-00843-f004:**
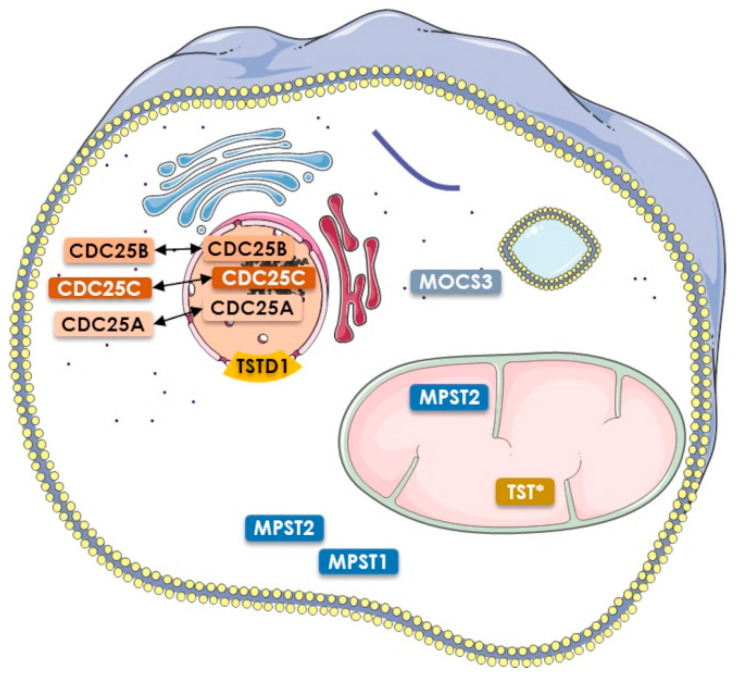
Reported sub-cellular localizations of human Rhd-containing proteins representatives [24,26,27,28,29,30,31,32,33,34].

**Figure 5 antioxidants-12-00843-f005:**
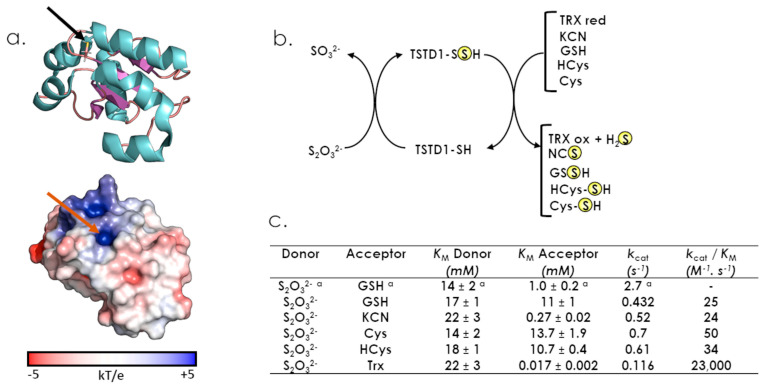
Structural and catalytic properties of TSTD1. (**a**) Cartoon and surface representations of the TSTD1 structure (PDB 6BEV). The active site cysteine is represented in stick and indicated by a black (cartoon) or a red (surface) arrow. (**b**) describes sulfur donors and acceptors for TSTD1. (**c**) Steady-state kinetic parameters for the thiosulfate sulfurtransferase activity of TSTD1 ([36], except for ^a^ [35]).

**Figure 6 antioxidants-12-00843-f006:**
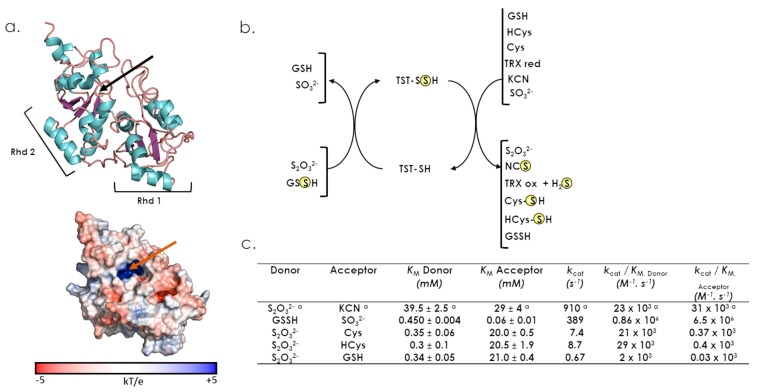
Structural and catalytic properties of TST. (**a**) Cartoon and surface representations of the TST structure (Alphafold AF-Q16762). The two Rhd domains are indicated. The active site cysteine is represented in stick and is indicated by a black (cartoon) or a red (surface) arrow. (**b**) describes sulfur donors or acceptors for TST. (**c**) Steady-state kinetic parameters for the sulfurtransferase activity of TST ([45], except for ^a^ [46]).

**Figure 7 antioxidants-12-00843-f007:**
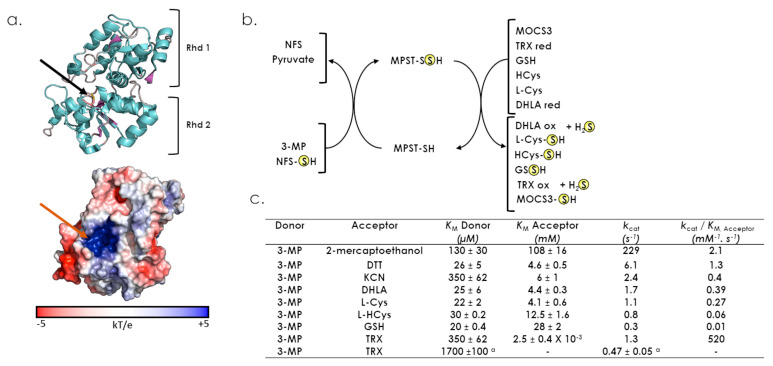
Structural and catalytic properties of MPST. (**a**) Cartoon and surface representations of the MPST structure (PDB 4JGT). The active site cysteine is represented in stick and indicated by a black (cartoon) or a red (surface) arrow. (**b**) describes sulfur donors or acceptors for the MPST. (**c**) Steady-state kinetic parameters for the 3-mercaptopyruvate sulfurtransferase activity of MPST ([54], except for ^a^ [25]).

**Figure 8 antioxidants-12-00843-f008:**
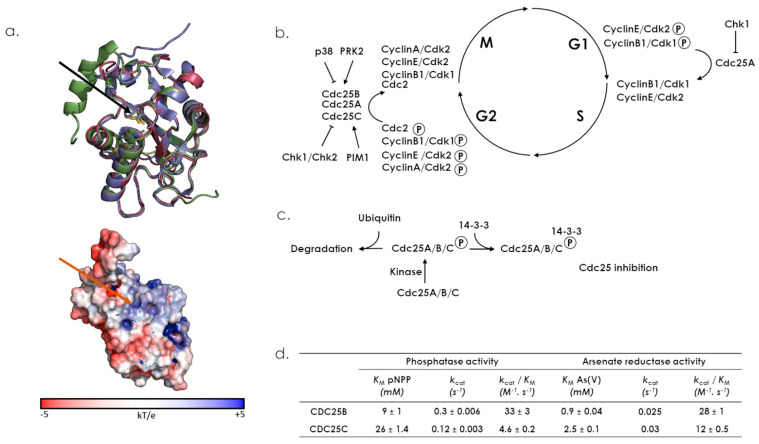
Structural features, catalytic properties and biological functions of CDC25s. (**a**) Upper panel: superimposition of Rhd domains 3D structures from CDC25A (1C25, red), CDC25B (3OP3, purple) and CDC25C (1CWR, green) using Pymol (version 2.4.1). RMSD values: 0.600 Å (1CRW versus 1C25) for 956 remaining atoms; 0.677 Å (1CRW versus 3OP3) for 951 remaining atoms; 0.913 Å (1C25 versus 3OP3) for 959 remaining atoms. Catalytic Cys are represented in sticks and indicated by a black arrow. Lower panel: surface charge representation of 1C25, position of the catalytic Cys is indicated by a red arrow. (**b**) CDC25 phosphatase activities toward Cyclin/Cdk complexes play key roles in cell cycle in regulating both G2/M and G1/S transitions. CDC25 activities are enhanced or inhibited by various kinases activities at different cell cycle phases. (**c**) CDC25 activities are also regulated by 14-3-3 reversible inhibition or proteasome activity. (**d**) Kinetic parameters of the phosphatase and arsenate reductase activities of purified Rhd domains of CDC25s [17]. Phosphatase activity was obtained from the rate of hydrolysis of p-nitrophenyl phosphate (pNPP), and arsenate reductase activity was determined using a coupled assay combining GSH, glutathione reductase, NADPH and glutaredoxine.

**Figure 9 antioxidants-12-00843-f009:**
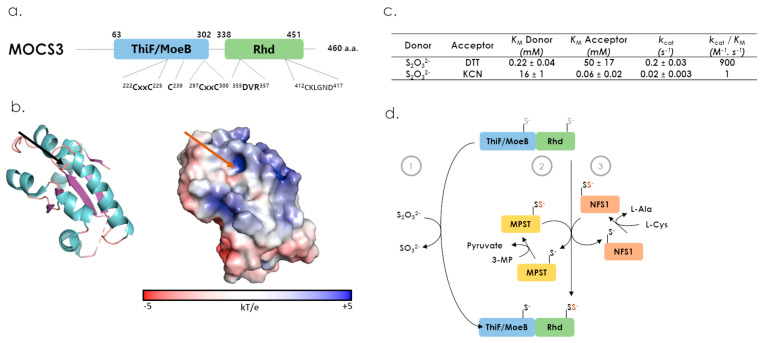
Structural features, catalytic properties and proposed physiological partners of MOCS3. (**a**) Modular organization of MOCS3 and conserved motifs in distant protein sequences identified by BlastP (Cys_222_XXCys_225_, Cys_239_, Cys_297_XXCys_300_, _355_DxR_357_ and _412_CxxGxD/R_417_). (**b**) Cartoon and surface representations of MOCS3-Rhd domain structure (PDB 3I2V). The active site cysteine is represented in stick and is indicated by a black (cartoon) or a red (surface) arrow. (**c**) Steady-state kinetic parameters for the thiosulfate sulfurtransferase activity of the Rhd domain from MOCS3 [79]. (**d**) describes sulfur donors for MOCS3 (1), MPST (2) or the cysteine desulfurase NFS1 (3).

**Figure 10 antioxidants-12-00843-f010:**
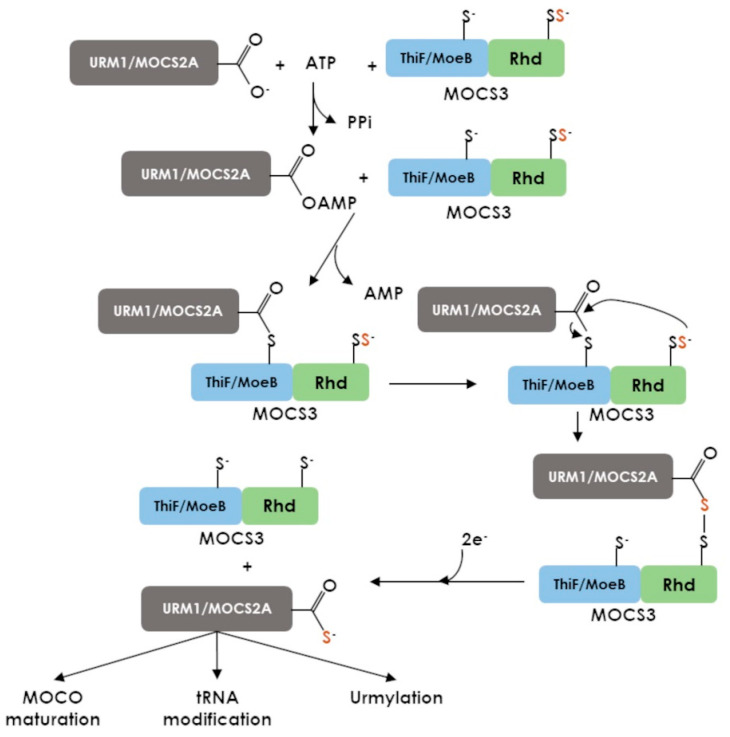
Proposed mechanism for the MOCS3-catalyzed reaction.

**Figure 11 antioxidants-12-00843-f011:**
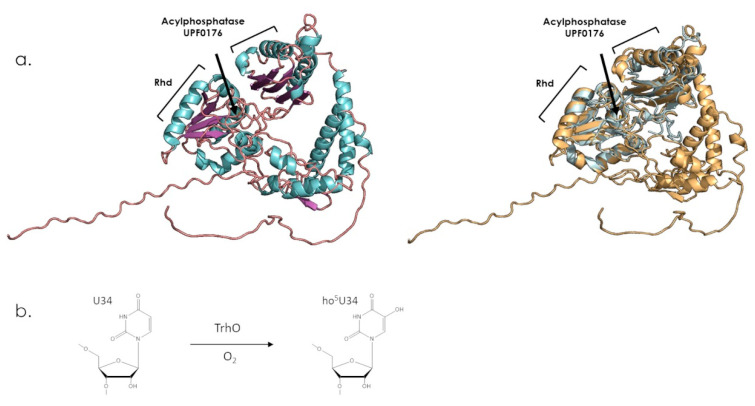
The TSTD2 structural model and reaction catalyzed by *E. coli* counterpart. (**a**) Left: the TSTD2 Alphafold predicted structure (AF-Q5T7W7). Acylphosphatase (UPF0176/IPR001792) and Rhd domains are indicated. Right: structure superimposition, using Pymol (version 2.4.1) (RMSD = 1.073 Å on remaining 1149 atoms), of the TSTD2 AlphaFold model (Gold) with the *Legionella pneumophila* protein structure (PDB 4F67, Pale green, Q5ZRP2, UniProt). Putative catalytic loop cysteine residue is represented as stick and is indicated by a black arrow. (**b**) The oxygen dependent tRNA U34 hydroxylation (^ho5^U34) catalysed by TrhO, the TSTD2 orthologous protein from *E. coli* [89].

## Data Availability

All data presented in this study are available in the article and Appendix A.

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
