# Peer review of "Rhodanese-Fold Containing Proteins in Humans: Not Just Key Players in Sulfur Trafficking"

_antioxidants, 2023, doi:10.3390/antiox12040843_

Round 1
Reviewer 1 Report
ALSOHAIBANI et al. wrote an authoritative and very informative review of the current understanding on active human rhodanese proteins, in both mechanistic and physiological terms. Overall, the manuscript is well written, very clearly organized, and comprehensive in terms of recent findings on structural and biochemical properties of human active rhodanese-containing proteins.
Minor concern
- Reference style is not consistent in the bibliography.
Author Response
We would like to thank the editor and all referees for their constructive comments and suggestions on our submitted manuscript entitled “Rhodanese-fold containing proteins in Human: Not just key players in sulfur trafficking” (antioxidants-2288036). Please find the detailed answers to the reviewer’s comments below.
Reviewer 1:
ALSOHAIBANI et al. wrote an authoritative and very informative review of the current understanding on active human rhodanese proteins, in both mechanistic and physiological terms. Overall, the manuscript is well written, very clearly organized, and comprehensive in terms of recent findings on structural and biochemical properties of human active rhodanese-containing proteins.
Minor concern
- Reference style is not consistent in the bibliography.
Thanks a lot for this recommendation, we have carefully checked reference formats and made appropriate modifications.
Reviewer 2 Report
The review of Alsohaibani et al. is about Rhodanese containing proteins in human. The review is very unique about this topics and is a good summary of the actual knowledge about this type of proteins. The review is quite well written however, it is full of small mistakes and unreadable details in the figures. The review must be proofread with the utmost care to eliminate them as much as possible and to reach a quality necessary for publication.
Title . Rhodanese domain ou fold containing proteins should be more adequate
L35. Change Nrf-2 by Nrf2
L50-51 and L59 are very similar. Repeat has to be removed
L68-69 : this information has to be shown on the figure
Figure 1. Panel a. Names of the structural domains are not readable. Panel B; The smallest letters are not readable.
All the figure caption. The panel a is always in bold instead of the title of the figure
L110. What means “the Human rhodanese equipment”?
L112, L338 to 345, L442: the font is changing
Figure 3. Panel C. Indicate the different sub-groups to improve the figure. Panel a. Specifiy the unit for the protein length.
L136. What is “TST* aka Rhodanese.
L139. Change “from structures superimpositions” to “from structure superimposition”
Figure 4. Why Rho protein?
L165. Specifiy what is HCys
Figure 5 . HCys is written HCy or Hcys… Panel C. What are the error bars?
Figure 6. The color bar is not readable. Panel b. What is the 2 below Chk1?
Figure 9. Panel C. Put the units under brackets. The color bar is not readable.
L528-529, 536-537, 538-540 : These parts are not fill in.
References. There is some diversities in the format of the references.
Author Response
We would like to thank the editor and all referees for their constructive comments and suggestions on our submitted manuscript entitled “Rhodanese-fold containing proteins in Human: Not just key players in sulfur trafficking” (antioxidants-2288036). Please find the detailed answers to the reviewer’s comments below.
Reviewer 2:
The review of Alsohaibani et al. is about Rhodanese containing proteins in human. The review is very unique about this topics and is a good summary of the actual knowledge about this type of proteins. The review is quite well written however, it is full of small mistakes and unreadable details in the figures. The review must be proofread with the utmost care to eliminate them as much as possible and to reach a quality necessary for publication.
Title . Rhodanese domain ou fold containing proteins should be more adequate
Title have been modified to “Rhodanese-fold containing proteins in Human: Not just key players in sulfur trafficking”
L35. Change Nrf-2 by Nrf2
Change have been done.
L50-51 and L59 are very similar. Repeat has to be removed
Thanks a lot for this point, lines 50-51 have been modified.
L68-69 : this information has to be shown on the figure
Thanks for this remark, but as these consensus sequences are not directly related to subclass organization, it is not possible to illustrate them in Figure 1C. In addition, detailed active site signatures are available for Human Rhd members in figure 3C.
Figure 1. Panel a. Names of the structural domains are not readable. Panel B; The smallest letters are not readable.
The figure 1A has been modified according to your recommendations. For the panel B we have increased the size of the weblogo, meanwhile some letters still not really readable. This cannot be improved since this is the rational of such representation indicating that the smallest letters correspond to poorly represented amino acids.
All the figure caption. The panel a is always in bold instead of the title of the figure
All captions have been modified with bold titles.
L110. What means “the Human rhodanese equipment”?
“Equipment” has been replaced by “repertoire”.
L112, L338 to 345, L442: the font is changing
Font has been homogenized.
Figure 3. Panel C. Indicate the different sub-groups to improve the figure. Panel a. Specifiy the unit for the protein length.
Modifications have been done.
L136. What is “TST* aka Rhodanese.
AKA means “also known as”, we have replaced the abbreviation by “also known as”.
L139. Change “from structures superimpositions” to “from structure superimposition”
Modification has been done.
Figure 4. Why Rho protein?
Sorry for this spell check error, we have replaced “Rho” by “Rhd”.
L165. Specifiy what is HCys
Done, in line 177 (new version)
Figure 5 . HCys is written HCy or Hcys… Panel C. What are the error bars?
We have done the homogenization for HCys and add error values from original papers.
Figure 6. The color bar is not readable. Panel b. What is the 2 below Chk1?
The color bar has been upgraded and Figure 8 panel b has been modified for Chk1/Chk2.
Figure 9. Panel C. Put the units under brackets. The color bar is not readable.
Done for all figures.
L528-529, 536-537, 538-540 : These parts are not fill in.
Modifications were done.
References. There is some diversities in the format of the references.
Thanks a lot for this point, we carefully checked reference formats and made modifications according to authors